# Addition of Natural Extracts with Antioxidant Function to Preserve the Quality of Meat Products

**DOI:** 10.3390/biom12101506

**Published:** 2022-10-18

**Authors:** Elisa Rafaela Bonadio Bellucci, Camila Vespúcio Bis-Souza, Rubén Domínguez, Roberto Bermúdez, Andrea Carla da Silva Barretto

**Affiliations:** 1Department of Food Technology and Engineering, UNESP—São Paulo State University, Street Cristóvão Colombo, 2265, São José do Rio Preto 15054-000, SP, Brazil; 2Centro Tecnológico de la Carne de Galicia, Avda. Galicia nº 4, Parque Tecnológico de Galicia, San Cibrao das Viñas, 32900 Ourense, Spain

**Keywords:** natural antioxidants, synthetic antioxidants, bioactive compounds, lipid oxidation, meat discoloration

## Abstract

Antioxidants are used to prevent oxidation reactions and inhibit the development of unwanted sensory characteristics that decrease the nutritional quality, acceptance, and shelf-life of processed meat products, improving their stability. Synthetic antioxidants, although efficient, are related to the development of diseases because they present toxic and carcinogenic effects. Thus, researchers and the meat industry are studying natural alternatives to synthetic antioxidants to be used in meat products, thus meeting the demand of consumers who seek foods without additives in their composition. These natural extracts have compounds that exert antioxidant activity in different meat products by different mechanisms. Thus, this review work aimed to gather studies that applied natural extracts derived from different plant sources as possible antioxidants in meat products and their action in preserving the quality of these products.

## 1. Introduction

Meat and meat products are affected by several factors during their shelf-life that compromise sensory acceptance by changing color, odor, flavor, and nutritional quality [1]. These changes can result in the rejection of the product and, due to the association with the freshness and safety of the product, discoloration negatively affects the purchase intention of consumers [2]. The main cause of deterioration is oxidation reactions. Thus, the use of antioxidants (synthetic or natural) is a means found by the industry to maintain the characteristics of fresh foods by inhibiting these oxidation reactions. Among the most used synthetic antioxidants are sodium erythorbate, sodium ascorbate, propyl gallate, butylated hydroxyanisole (BHA), tert-butylhydroquinone (TBHQ), butylated hydroxytoluene (BHT), and curing salts such as nitrite and nitrate, all of which are associated with toxicological and carcinogenic effects [3].

It is evident that consumers are concerned about the use of chemical additives mainly due to their carcinogenicity, which results in an increased demand for clean-label foods. In this way, the risk associated with synthetic antioxidants added to this change in habit prompted the industry to look for natural additives [4]. Consequently, studies are being carried out in order to find natural alternatives for synthetic additives. Among the possibilities found is the use of plant extracts in meat products, for which their action in maintaining sensory and nutritional quality has been evaluated. These extracts contain high levels of phenolic compounds and other antioxidant substances effective in delaying oxidative reactions [3,5,6]. Thus, this review work aimed to gather studies that applied natural extracts derived from different matrices, such as fruits and spices, in addition to essential oils and residues and by-products of the industry, as possible antioxidants in meat products and their action in preserving quality during shelf-life (Figure 1).

## 2. Oxidation Reactions in Meat and Meat Products

Oxidation reactions are the main non-microbial causes of meat deterioration during the storage period [2,7], reducing its quality (changes in texture, formation of a characteristic flavor of a rancid product, and manly discoloration). The presence of unsaturated fatty acids and an increase in oxygen exposure through process steps such as milling, slicing, and cooking intensifies these reactions [6]. In addition, these reactions decrease nutritional quality due to the loss of essential fatty acids and vitamins. However, the most impacting consequence refers to the formation of toxic products during lipid oxidation, which are considered harmful to health because they are associated with pathologies such as atherosclerosis, cancer, and inflammation [3].

The lipid oxidation process is complex and has many mechanisms that interact with each other [8]. In simple terms, oxidation consists of the loss of one or more electrons by the molecule, most characterized by the loss of hydrogen and gain of oxygen [9]. According to Lorenzo et al. [6], oxygen is the main factor that affects lipid oxidation, reacting with unsaturated fatty acids and producing peroxides. The oxidation process occurs in three stages called initiation, propagation, and termination [3,6], and results in the modification of fatty acids through a self-catalytic mechanism called self-oxidation [10].

Initiation occurs through the action of pro-oxidizing agents or reactive oxygen species (ROS) and conditions favorable to the reaction such as the thermal process and presence of light, and forming a fatty acid radical (R •) through the removal of the hydrogen radical. In the second stage, the propagation phase, the formation occurs of peroxide radicals (ROO •) through the reaction between the radical previously formed in the initiation step and the molecular oxygen (O_2_). In the next reaction, primary products are formed, the so-called hydroperoxides, which are considered relatively stable products [10]. Peroxide and hydroperoxide radicals have no taste or odor. However, metal ions, light, and heat cause isomerization and decomposition reactions of hydroperoxides, giving rise to secondary products such as hexanal, pentanal, 4-hydroxynonenal, and malonaldehyde (MDA), which are responsible for the odor, flavor, and texture characteristic of rancidity. Finally, stable or non-reactive products are formed in the termination phase from various reaction combinations that occur between free radicals or from the reaction of free radicals with non-radical compounds such as antioxidants [3].

The most important aldehyde produced during lipid oxidation is malonaldehyde (1,3-propanedialdehyde) because it can produce a rancid aroma even in small quantities and it is used to quantify lipid oxidation through analytical methods in meat and meat products. The main method of analysis used to quantify MDA is the thiobarbituric acid (TBA) test, which consists of the colorimetric measurement of the complex formed between TBA and MDA. Some studies note that values of lipid oxidation between 2 and 2.5 mg MDA/kg are considered accepted limits and still do not affect the sensory acceptability of the product [11,12,13], but other more restrictive studies indicate that rancid flavor could be detected at values about 0.6 mg MDA/kg [14,15].

Many components of the meat can be affected by oxidation reactions, among them proteins which, being “attacked” by reactive oxygen species, lose the sulfhydryl groups, generating carbonyl compounds [16]. The formation of carbonyl compounds is considered the main change in the oxidized protein. There are four ways that carbonyls can form from protein oxidation: the first is when the amino acid side chain is directly oxidized; the second is peptide structure fragmentation; the third is when protein and reducing sugars react with each other; and, finally, when a bond between protein and non-protein carbonyl compounds occurs.

Protein contributes to important technological, nutritional, and sensory properties, so is considered the main constituent of meat. Thus, modification of its structure results in protein denaturation or proteolysis. Due to protein oxidation, meat systems become more susceptible to the action of proteolytic enzymes, in addition to undergoing polymerization of proteins, producing soluble aggregates that can cause gelation and emulsification, or insoluble aggregates that prevent binding with water, affecting its solubility [17].

Finally, the impact caused by protein oxidation that most affects the quality of meat and meat products is color change, more specifically called pigment oxidation [18]. Myoglobin, the meat pigment, is a protein (globin) linked to a non-protein prosthetic group (heme) that contains an iron molecule. The color variation of this pigment depends on the oxidative state of the iron present in the heme group. Myoglobin is purplish red in color; however, it becomes bright red in the presence of high oxygen concentration, as it becomes oxymyoglobin and the iron ion is reduced (Fe^++^). It is concluded that the partial pressure of O_2_ is responsible for the oxidation states of myoglobin and that a low partial pressure favors the formation of deoxymyoglobin, an unstable pigment. In the presence of oxygen and through the action of free radicals, deoxymyoglobin becomes metmyoglobin, which is brown in color and iron is in oxidized form (Fe^+++^) [3,6].

Therefore, commercially, with color being the sensory attribute that most influences the consumer’s purchase intention [19], oxidation reactions increase the possibility of product rejection. Therefore, the shelf-life includes the moment when the consumer can detect rancid characteristics (mainly volatile) or observe changes in the color caused by oxidation of meat and meat products.

Antioxidants are used by the industry as the main strategy to decrease the intensity of oxidation reactions in meats and their processed derivatives, increasing the useful life of these products [20]. Synthetic compounds with phenolic structure capable of acting through the elimination of peroxyl radicals or canceling the formation of free radicals are used for this purpose [6]. The reaction of antioxidants with oxidation is believed to occur by donating electrons to break and terminate the oxidation at the propagation step, thereby preventing additional lipid and protein radicals from forming [21].

The mechanism of action of natural antioxidants is different depending on the compound. In general, the antioxidant effect is associated with their capability of eliminating reactive species, their curbing of autoxidation by donating a hydrogen atom to stabilize the first oxidation products, their potential chelation of metal ions (pro-oxidants), and their role as an inhibitor of the hydroperoxide’s degradation [4]. Phenolic compounds are the main natural antioxidants; among the most important are tocopherols, flavonoids, and phenolic acids. These compounds have a strong ability to donate hydrogen and free radical scavenging capability [4,10]. According to Oswell et al. [9], the compounds that have antioxidant activity in meat products are carotenoids, hydroxycinnamic acid, flavonoids, terpenes, and antioxidant vitamins.

## 3. Antioxidant Substances Derived from Fruits Applied in Meat Products

Studies using fruit or extracts derived from fruits as a source of antioxidant compounds have been widely carried out in meat products, as many works have been published in recent years [7,19,22]. Fruits are rich sources of bioactive compounds and pigments such as phenolic compounds, including anthocyanins and catechins, in addition to carotenes and betacyanin, which have high antioxidant capacity.

Some works evaluated pink guava fruit as an antioxidant in meat products. In this sense, Chang et al. [23] observed that the flesh of this fruit presented important antioxidant activity and contained a large amount of vitamin C and phytochemicals such as lycopene, carotenoid, and flavonoids. Moreover, phenolic compounds such as anthocyanins, apigenin, myricetin, and ellagic acid contribute ascorbic acid to its antioxidant activity. Therefore, pink guava pulp was evaluated as an antioxidant by Joseph et al. [24] in raw pork emulsion for nine days in refrigerated storage and aerobic packaging. The authors found that guava pulp improved the color stability during storage and stated that the increase in added concentration increased *a** values due to the lycopene present in the emulsion (1.9 vs. 4.4 mg/kg in 0 day and 1.7 vs. 3.5 mg/kg in nine storages for 5 vs. 10%). They also observed that the pink guava pulp was able to minimize lipid oxidation, showing higher values of TBARS for the control without the addition of pulp, and the values were reduced according to the increase in the added concentration of the pink guava pulp. Pink guava pulp was also studied, in addition to tomato-derived products (tomato puree, freeze-dried tomato peel, and tomato pulp), by Joseph et al. [25] in pork emulsified products (Table 1). In this study, the tested antioxidants improved the color, minimized the formation of pigment metmyoglobin, and improved oxidative stability in raw meat emulsion, at the concentrations studied.

By comparison, pomegranate is a fruit recognized for its health benefits due to the presence of bioactive compounds in its composition. In the fruit, peel corresponds to approximately 50% of its total weight and, being rich in phenolic compounds, it is a valuable residue of the industry. Among the phenolic compounds found in pomegranate peel are flavonoids (catechins, anthocyanins), hydrolysable tannins, and phenolic and organic acids. These compounds characterize it as an important antioxidant, antimicrobial, antifungal, and pigment [31].

Grape seed can be used as an alternative to synthetic antioxidants due to its high content of polyphenolic compounds and proanthocyanidins. Kulkarni et al. [29], for example, studied the addition of grape seed extract to a pre-cooked, frozen, and re-heated beef sausage model system and its effect on lipid, color, and sensorial properties. In this study, the concentrations of extract tested were 100, 300, and 500 mg/kg and it was compared to three other treatments: ascorbic acid (100 mg/kg of fat), propyl gallate (100 mg/kg of fat), and a control (without antioxidants). The authors found that the extract used at the lowest concentrations showed similar results to propyl gallate for oxidative, color, and sensory stability in 4 months of storage. In pork liver pate, the addition of grape seed extract (1000 mg/kg) improved the oxidative stability more than the treatment added with BHT (200 mg/kg) during storage at 4 °C for 24 weeks, confirming its antioxidant capacity in meat products [32].

In pork patties, the addition of guarana seed extract and its effects was evaluated by Pateiro, et al. [30]. In this work, the patties were kept under refrigeration (2 ± 1 °C) and analyzed for 18 days (Table 1), and it was observed that the guarana seed extract at both concentrations (250 and 500 mg/kg) not only reduced lipid oxidation when compared to BHT (200 mg/kg), but also improved color, lipid, and protein stability. It should also be noted that the guarana seed extract has many phenolic compounds, with tyrosol and other low molecular weight phenolics appearing in greater quantities.

The stability of the beef patty during storage (six days under refrigeration and retail conditions) was tested through an interesting approach. To evaluate the effect of a new combination of extracts, Fruet et al. [33] developed an antioxidant with lemon, orange, and buffered vinegar and compared it with acerola and rosemary extracts in beef patties. Although rosemary and acerola extract (0.30 and 0.25%) were similar in relation to antioxidant activity, there was an increase in lipid oxidative products during storage. However, the proposed extract reduced lipid oxidation in the concentration used (0.6%), inhibiting the formation of oxidation products and, consequently, inhibiting the increase in TBARS values during storage. Moreover, it efficiently inhibited the growth of aerobic bacteria during storage due to the buffered vinegar in this extract.

Betacyanin as betanin is present in pitaya, which is known not only for its color but also for its antioxidant activity, which is up to double that of some anthocyanins. According to Cunha et al. [7], pitaya has a large amount of beta-carotene, betacyanins, lycopene, vitamin E, vitamin C, and polyphenols. Within this context, a study was developed to investigate the antioxidant effect of red pitaya extract in pork patties with the replacement of animal fat by a tiger nut oil emulsion [27]. In this work, an aqueous extract and lyophilized red pitaya pulp (250, 500, and 1000 mg/kg) were added to the patties as a natural antioxidant. A control (without antioxidant) and a treatment of sodium erythorbate (500 mg/kg) were examined. The authors observed that red pitaya was able to reduce discoloration over time and reduce lipid oxidation reactions and carbonyl formation when compared to the control. These results are expected, since several plant extracts protect meat products from the main degradative reactions, such as oxidative processes [34].

The extract obtained from açai palm fruit (*Euterpe oleracea* Mart.) has been studied for application in meat products. This fruit naturally occurs in the Amazon region and is widely known for having high antioxidant activity in vitro due to its rich composition of phenolic compounds, such as different anthocyanins, phenolic acids, and flavones [35]. Therefore, Bellucci et al. [22] applied an extract obtained from açaí pulp in pork patty to evaluate its antioxidant activity (Table 1). Açaí extract reduced lipid oxidation in pork hamburger as much as sodium erythorbate.

However, bioactive compounds derived from plants are unstable chemically and can undergo oxidative degradation by exposure to oxygen, light, metal ions, pH, and other parameters, and their direct incorporation in food is very restricted [36]. In this regard, microencapsulation is an innovative technique that can be used to preserve the bioactive compounds present in plant extracts, enabling greater stability and preserving their functional quality [7]. Studies were performed to investigate the preservation of bioactive compounds by microencapsulation [37].

In this regard, Cunha et al. [7] evaluated the action of microencapsulated extract from pitaya peels on pork hamburgers submitted to a high-pressure process. The process causes protein denaturation and cellular membrane damage, releasing non-heme iron, and consequently increasing lipid oxidation. The authors reported that the microencapsulated pitaya peel extract (100 and 1000 mg/kg), despite reducing lipid oxidation as much as BHT (100 mg/kg), was able to minimize it and reduce protein oxidation after 9 days of refrigerated storage in an emulsified pork product subjected to a high-pressure process. Moreover, the extract rich in anthocyanin (cyanidin-3-o-glucoside) obtained from Jaboticaba peel was microencapsulated by a spray-dryer into maltodextrin to be applied in fresh pork sausage as a natural antioxidant [28,38]. In this work, microencapsulated jaboticaba extract was added at the concentrations of 2 and 4%, and reduced lipid oxidation during 15 days of refrigerated storage, presenting TBARS values lower than the other treatments (cochineal carmine and the control) [28]. In addition, in this study, the extract (2%) did not affect the overall acceptance of fresh sausage, showing itself to be an alternative as a natural coloring agent.

## 4. Spices and Essential Oils Applied in Meat Products as Naturals Antioxidants

Herbs and spices, or product derivates from them, have been used to preserve food due to their antioxidant properties, mainly in meat products. In addition, its use is beneficial to prevent oxidation reactions and to develop clean label products, which are desirable for consumers [39]. Many studies have been carried out to evaluate its effect on the stability of lipid oxidation, protein, and color in meat products. For example, the aqueous extract of Dog rose *(Rosa canina* L., RC) was tested by Vossen et al. [40] in porcine sausage (frankfurter) to determine relevant quantities of phenolic compounds (Table 2), such as catechins and procyanidins, and ascorbic acid, in its composition. The concentrations of 5 and 30 g/kg of extract were compared to both a positive control treatment added to ascorbic acid (0.5 g/kg) and sodium nitrite (0.1 g/kg), and a negative control without the addition of any antioxidant. In this study, dog rose extract decreased the generation of products from lipid oxidation as much as the combination of sodium nitrite and ascorbic acid; however, it was shown to be more efficient in inhibiting protein oxidation during 60 days at 2 °C. Moreover, the authors reported a possible contribution of the extract in the pink sausage coloration.

The antioxidant effect of rosemary extract has been demonstrated in many studies in different meat products such as fresh and cooked sausages [33]. Doolaege et al. [48], for example, investigated the action of different concentrations of rosemary extract (0, 250, 500, and 750 mg/kg) on lipid oxidation and color stability in liver pâté with sodium nitrite at concentrations of 40, 80, and 120 mg/kg. Rosemary extract contains phenolic diterpenes (carnosol and carnosic acid) that act as hydrogenic donors in the chain reaction of free radicals, and its antioxidant action is well-known. In this study, the addition of the extract provided lower values of TBARS, and therefore decreased lipid oxidation in raw and cooked liver pâté; however, no concentration effect was observed between 250 and 750 mg/kg. At the same time, high concentrations of antioxidants ascorbic acid, α-tocopherol, and carnosic acid were maintained.

Turmeric *(Curcuma longa* L.) is widely used as a seasoning, dye, and preservative in food in Southeast Asia, India, and China. Several sesquiterpenes and curcuminoids were isolated from the turmeric rhizome, and curcuminoids were responsible for their yellow color action, and equivalent between 2 and 7% of their composition [41]. The condiment is obtained by boiling the roots, followed by drying; however, for the study developed by de Carvalho et al. [41], the extract was obtained by supercritical fluid and presented high antioxidant capacity from the ABTS and DPPH assays (1490.53 mg ascorbic acid/100 g and 42.92 mg Trolox/g, respectively). When applied to lamb sausage with fat substitution by vegetable oil emulsion, the extract at the concentrations used (250, 500, and 750 mg/kg) had better results in preventing lipid oxidation than sodium erythorbate (500 mg/kg) during the 18 days of refrigerated storage.

Pink pepper extract *(Schinus terebinthifolius* Raddi) had an antioxidant effect as much as that of BHT in chicken burgers for 7 days under refrigeration [43]. The authors state that pink pepper extract, which contains ascorbic acid, carotenoid, and phenolic compounds, is a good and effective alternative as an antioxidant to be used in a meat product without affecting sensory attributes.

Clove is another widely studied spice as it is an aromatic condiment used in food and is added to increase shelf-life and reduce food deterioration. Clove contains sesquiterpenes, triterpenoids, and eugenol (4-alil-2-methoxyphenol), and its main bioactive compound has insecticidal and antioxidant properties [49]. After obtaining clove extract using many solvents, El-Maati et al. [49] concluded that ethanol is more effective for extraction of phenolic and flavonoids, and the antioxidant capacity is directly proportional to the amount of total phenolic compounds.

Zahid et al. [44], however, used water as a solvent to obtain clove extract and applied it to cooked beef patties stored under refrigeration. After 10 days of storage, the aqueous extract of clove (0.1 g/100 g) showed the best antioxidant activity, as it had lower values of lipid and protein oxidation compared to BHT (0.02 g/100 g). Regarding ascorbic acid at 0.05 g/100 g, the clove extract had higher values of *a** and color notes in the sensory analysis, preserving the color of hamburgers, in addition to lower TBARS values. All antioxidants used reduced lipid and protein oxidation, maintained better *a** values, and preserved the stability of sensory attributes when compared to the control without added antioxidants.

Essential oils are incorporated in meat products as an alternative to preserve oxidation reactions because they are composed of many bioactive compounds, including terpenes, terpenoids, and phenolics, which have proven antimicrobial, antifungal, antioxidant, antiviral, antimycotic, antiparasitic, and insecticide properties [50]. In addition, essential oils are classified as Generally Recognized as Safe (GRAS); that is, they are safe and allowed for use as additives in foods. Moreover, they are widely accepted by consumers, generating greater interest in their application in meat products. Aromatic plants are the main sources of essential oils, for example, rosemary, oregano, sage, ginger thyme, and mint.

Many studies have been conducted to investigate the action potential of various essential oils to inhibit oxidative reactions and the growth of microorganisms in products [47,51,52]. The compounds responsible for antioxidant activity present in essential oils are terpenes, phenylpropanoids, alcohols, aldehydes, and ketones, while compounds such as thymol, cymene p, γ-terpinene, and carvacrol are related to their antimicrobial activity and may act against *Clostridium s**porogeneses*, *Staphylococcus aureus*, *Escherichia coli*, *Enterococcus faecalis*, and *Clostridium perfringens* in various meat products.

Estévez and Cava [45] applied rosemary essential oil in frankfurters containing Iberian pig and white pig, and evaluated this effect as an antioxidant against oxidation reaction. In white pig frankfurters, 300 and 600 ppm reduced TBARS values, and the highest level also reduced hexanal formation. However, rosemary essential oil in low concentration (150 ppm) reduced lipid and protein oxidation, and the higher concentrations studied (300 and 600 ppm) favored the oxidation reaction in Iberian pig frankfurters. Thus, the authors reported that the concentration of the natural antioxidant used and the components of the meat influence oxidative stability, and some essential oils can act as pro-oxidants when added at high concentrations.

Spice essential oils were studied as a preservative in meat products by Sharma et al. [47]. The essential oils obtained from clove, cassia, holy basil, and thyme were incorporated into fresh chicken sausages packed in vacuum packaging stored for 45 days at 18 ± 2 °C. According to the authors, clove oil presented the lowest values of TBARS followed by cassia oil; however, all essential oils were effective against lipidic oxidation when compared to the control. Regarding antimicrobial activity, cassia oil showed the lowest values in the total count of microorganisms, psychotropic and molds, and yeasts. Sensorially, the essential oils added to chicken fresh sausage protected against discoloration during storage.

## 5. Flowers, Leaves, and Vegetable Residues as Natural Sources of Antioxidants Applied in Meat Products

Extracts of leaves, flowers, and other plant residues rich in bioactive compounds are being studied as a preservative in meat products due to their antioxidant properties, and are able to reduce or prevent oxidation reactions. Olive leaves, for example, are by-products of olive oil production and are obtained in large quantities during tree pruning, harvesting, and in olive oil industries. These by-products are rich in important bioactive compounds such as polyphenols, which present antioxidant, hypoglycemic, and anti-inflammatory properties, as supported by scientific evidence. In this connection, Khemakhem et al. [53] added olive leaf extract and hydrolyzed olive leaf extract (Table 3) at a concentration equivalent to 200 mg gallic acid to each kilogram of salmon to produce a salmon burger, which the authors claimed to be sufficient for antioxidant activity. The extracts extended the shelf-life of the salmon burger, as they not only decreased the formation of volatile nitrogen bases and trimethylamine (indicators of deterioration in fish), but also reduced lipid oxidation during refrigerated storage (4 °C) when compared to the control.

In another study, Lorenzo et al. [5] evaluated the effect of pitanga leaf extract (250, 500, and 1000 mg/kg) on lipid and protein oxidation in pork burgers. The main phenolic compounds found in the extract were hydroxycinnamic acid (cinnamic and caffeic acids were the most abundant), tyrosol, alkylmethoxyphenols, hydroxycoumarins, and hydroxyphenylpropenes, which are responsible for the high antioxidant activity in in vitro assay, resulting in a similar behavior to that of BHT (200 mg/kg) in relation to lipid and protein oxidation. As expected, pitanga leaf extract and BHT maintained TBARS values below 0.5 mg MDA/kg during all storage times. The same did not happen with the control (without antioxidants), which exceeded this value on the seventh day of refrigerated storage. In a different study, the pitanga leaf extract at 250 mg/kg was more efficient against oxidation reactions (lipid and protein) than the other antioxidants used in lamb burger (guarana seed extract at 250 mg/kg and BHT at 200 mg/kg) [1].

Extracts of thyme by-products, with high amounts of flavonoids, terpenoids, and tannins, among other bioactive compounds, obtained through sustainable extraction by supercritical fluid, were added to ground pork patties by Šojić et al. [54]. The authors obtained two extracts through supercritical fluid extraction in different conditions. In this study, two pressures (100 and 350 bar) and extraction temperatures (40 °C and 50 °C), using only CO_2_ as solvent, were used to compare the effect of extraction conditions on the chemical constituents and bioactivity of the extracts. The main terpenoids of both were thymol, α-terpineol, and carvacrol. The extracts were added in concentrations ranging from 0.075 to 0.150 μL/g (four treatments) to patties. The control (without antioxidant) presented values above 0.5 mg MDA/kg for lipid oxidation from the second day of storage, while the patties to which thyme by-product extract was added did not reach this TBARS value, even on the third day. In addition, the extracts avoided discoloration of the samples, confirming their use as an antioxidant in ground pork patties.

Three different industrial residues described as sources of phenolic compounds and with wide antioxidant capacity were studied by Munekata et al. [55]. The extracts were obtained from the residues of beer production, chestnut leaves, and peanut skin discarded during processing. These extracts were applied at the concentration of 2000 mg/kg in the fermented product (Spanish Salchichón) containing encapsulated *n*-3 long chain fatty acids. There was no difference among the treatments, including the control, in the analysis of lipid oxidation by TBARS due to the protection of fatty acids generated by encapsulation. However, BHT and beer residue extract reduced protein oxidation, and all antioxidants tested (synthetic and natural) were able to reduce the formation of aldehyde compounds in comparison to the control.

In this context, Rodrigues et al. [56] investigated the antioxidant activity of the different parts of banana inflorescence and found that the extract obtained from male flowers presented the highest values for total phenolics and flavonoids. It also presented the highest antioxidant activity (FRAP) and lower CI_50_ (half of the maximum inhibitory concentration determined by the DPPH method). The banana male flower extract was tested in concentrations of 0, 0.5, 1, 1.5, and 2% in raw sausage storage under refrigerated conditions (4 °C) for 28 days. The authors reported that the concentrations used did not compromise the sensory acceptance and the color of the product, and were also effective in keeping TBARS values lower than the control.

## 6. Other Sources of Natural Antioxidants Applied in Meat Products

Macroalgae and microalgae, present in abundance in the marine environment, can be considered an important natural source of antioxidants due to the pigments present in their cells, such as carotenoids, chlorophylls, and other compounds including polyphenols, alkaloids, tocopherol, and terpenes [2,57]. *Haematococcus pluvialis* is rich in a red-purple carotenoid known as astaxanthin, which has the highest antioxidant capacity among the natural substances. In this context, *Haematococcus*
*pluvial* extract containing 5% astaxanthin was added by Pogorzelska et al. to ground pork meat at concentrations of 0.15, 0.30, and 0.45 g/kg to evaluate its effect on oxidative and color stability of this product during refrigerated storage [58]. The extract studied showed high free radical inactivation capacity and, mainly, inhibited lipid oxidation at concentrations of 0.3 and 0.45 g/kg, increasing the oxidative stability of pig hamburger. At all concentrations, it was effective in terms of color stability, presenting the highest values of *a**, thus favoring product acceptance. According to the authors, *Haematococcus*
*pluvialis* seaweed extract can be applied to meat products as an antioxidant.

In pork patties formulated with *oleogels*, *Fucus vesiculosus* extract was added. This extract is a brown macroalgae rich in a phenolic compound (phlorotannin) known to have higher antioxidant capacity than commercial antioxidants [2]. The extract added at a concentration of 1000 mg/kg promoted higher oxidative stability (lipid and protein) in pork patties than the other extract concentrations tested (250 and 500 mg/kg) and the control, but did not show better results than the synthetic antioxidant BHT (200 mg/kg). However, differences were not conserved between the results in sensory analysis, and the extract at 500 mg/kg promoted better scores for the odor attribute. For the authors, the benefits presented by *Fucus*
*vesiculosus* extract were limited for later application in meat products, and techniques should be studied to expand its antioxidant activity.

In recent studies, thiamine and non-specific lipid-transfer protein isolates (TdLTP4 protein) were identified as potential antimicrobials and antioxidants, which could be used to improve food preservation and human health [59,60]. In fact, increasing amounts of thiamine (0.1 and 0.5 g/100 g) added to minced beef meat showed better physicochemical and microbiological parameters, while reformulated batches had significantly lower TBAR values than the control and samples formulated with synthetic antioxidant (BHT). This corroborates the potential use of this amino acid as a natural food preservative in the meat industry [59].

## 7. Final Considerations

Natural extracts, vegetable powders or essential oils, industrial waste, and different natural sources can extend the shelf-life of meat products. The antioxidant capacity results from bioactive compounds present in these vegetables, such as phenolic compounds including flavonoids, phenolics acid, terpenes, and terpenoids, or compounds that confer colors, such as curcuminoids, carotenoids, and betacyanins, which are found in barks, seeds, leaves, and residues and by-products of vegetables, in addition to other non-plant sources, such as seaweed. Their main action to retain the freshness characteristics of meat products for longer is against lipid and protein oxidation, and against the discoloration of these products. Avoiding discoloration for longer is important to retain the product’s attractiveness throughout its shelf-life because color directly influences the consumer’s purchase decision. Finally, the use of natural extracts in foods meets the new eating demands of consumers, who have increased their concern about health and are prioritizing the consumption of more natural foods with fewer preservatives.

## Figures and Tables

**Figure 1 biomolecules-12-01506-f001:**
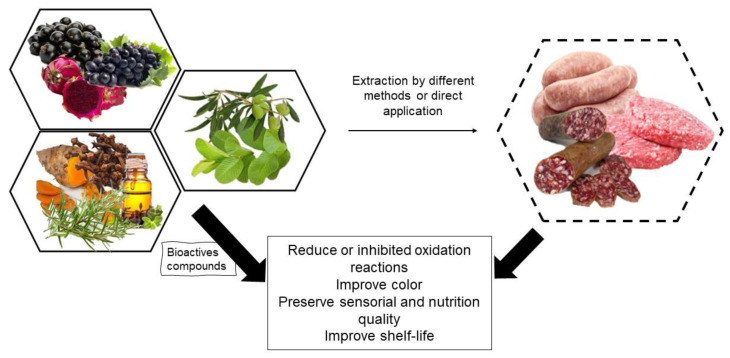
Illustration representing the application of natural sources as an antioxidant in meat products as a means of preserving quality and shelf-life.

**Table 1 biomolecules-12-01506-t001:** Fruit (pulp, pigment, peel, and seeds) as a source of antioxidants applied in meat products.

Source or Active Compound	Levels Added (%)	Adding Methods	Sample	Storage Conditions	Effect	Ref.
Pink guava pulp (PGP) (β-carotene and lycopene)	5, 7.5 and 10	Paste from pink guava pulp	Raw pork emulsion	Nine days in refrigerated storage and aerobic packaging	Increase redness (improves color) Reduces metmyoglobin formation High inhibition of lipid oxidation at 10%	[24]
Tomato products and pink guava pulp (PGP) (β-carotene, lycopene, and others)	Tomato puree—10%; tomato pulp—12.5%; lyophilized tomato peel—6% and PGP—10%	Tomato puree, paste of tomato pulp; powder tomato peel and pink guava pulp	Pork emulsion	9 days at 4 ± 1 °C in darkness and aerobic conditions	Improves color and reduces lipid oxidation	[25]
Plum (*Prunus salicina*) peel and pulp microparticles (proanthocyanidins and cyanidins)	2.0% *w*/*w* level	Powder extract	Breast chicken patties	Polyethylene films (oxygen permeability = 5.20 × 10^−12^ m^3^·m^−2^·s^−1^·Pa^−1^); dark, at 4.0 ± 0.5 °C for 10 days	Improve color end reduced lipid oxidation by around 50% during 10-days storage	[26]
Red pitaya extract (Polyphenols and betacyanin)	0.05, 0.07 and 0.10	Powder extract	Pork patties with replacement of animal fat	Modified atmosphere (80% O_2_ and 20% CO_2_); fluorescent light for 18 days at 2 ± 1 °C	Cooking loss and texture profile was not compromised by red pitaya extract showing no difference compared to the control and sodium erythorbate treatment; increased redness and improved color stability; reduced lipid oxidation and improved sensorial scores	[27]
Açaí extract powder (Anthocyanins and proanthocyanidins)	0.025, 0.05 and 0.075	Powder extract	Pork patty	Packed in nylon-polyethylene bags without vacuum for 10 days in the dark at 2 °C	pH and cooking parameters (weight loss and shrinkage) were similar between the treatments. The extract reduced lipid oxidation and 0.05 and 0.075% compromised color parameters; 0.025% was effective as natural antioxidant	[22]
Microencapsulated extract of pitaya (*Hylocereus c*.) peel (β-carotene, betacyanin, lycopene and polyphenols)	0.01 and 0.10	Powder extract	Pork patty submitted to high-pressure processing	9 days of storage at 4 °C under aerobic packaging	Able to reduce protein oxidation after 9 days of storage	[7]
Microencapsulated jaboticaba (*Myrciaria cauliflora*) extract (Anthocyanins)	2 and 4	Powder extract	Fresh pork sausage	Dark and aerobic condition for 15 days under refrigeration (1 ± 1 °C)	Reduce lipid oxidation; in sensory acceptance, 4% decreased the scores of texture, color, and overall acceptance of the pork sausages; 2% presented scores similar to the control and carmine treatment for major sensorial attributes.	[28]
Grape seed extract (Resveratrol)	0.01, 0.03, and 0.05	Powder extract	Beef sausage model system	Package in PVC; frozen at −18 °C for 4 months	All concentrations protected the meat product model system against lipid oxidation	[29]
Guarana seeds extract (Catechins, epicatechins and proanthocyanidins)	0.025, 0.05 and 0.10	Powder extract	Pork patty	Modified atmosphere (80% O_2_ and 20% CO_2_); under light for 18 days at 2 ± 1 °C	Shown higher inhibitory effect on oxidation reactions (lipidic and protein) than BHT at 0.025, 0.05%; improvement of product color	[30]

**Table 2 biomolecules-12-01506-t002:** Spices, their derived products, and essential oil as natural sources of antioxidants applied in meat products.

Source or Active Compound	Levels Added (%)	Adding Method	Sample	Storage Conditions	Effect	Ref.
Dog rose (*R. canina* L.) extract (Procyanidins, catechins and ascorbic acid)	0.5 and 3.0	Liquid extract	Porcine sausage (frankfurter)	Oxygen permeable poly-vinyl chloride film; dispensed in polypropylene trays; stored for 60 days at 2 °C in the dark	Both concentrations inhibited lipid and protein oxidation	[40]
Turmeric (*Curcuma longa* L.) extract (curcuminoids)	0.025, 0.05 and 0.075	Liquid extract	Fresh lamb sausage with fat replacement by Tiger nut (*Cyperus esculentus* L.) oil	The package was under a modified atmosphere (80% O_2_ and 20% CO_2_) in 300 mm thick PET-EVOH-PE trays, sealed with multilayer PE-EVOH-PE film, and stored at 2 ± 1 °C under light for 18 days.	0.075% showed the highest antioxidant activity throughout the storage period; reduced lipid oxidation;	[41]
Turmeric (curcuminoids—curcumin, desmethoxycurcumin, and bisdemethoxycurcumin) and black pepper spices (piperine)	Turmeric powder—ranged from 0 to 2.4% and black pepper—ranged from 0 to 0.44%	Powder	Cooked meat patties	Stored at −20 °C until tested.	A decrease in lipid peroxidation.	[42]
Pink pepper extract (Ascorbic acid, carotenoids and phenolics compounds)	0.739% (equivalent to 90 mg GAE/kg meat)	Liquid extract	Chicken burger	Two different conditions: aerobic packing (PVC film) and vacuum; stored at 2 °C with white light incidence; evaluated for consecutive 7 days	Pink pepper extract reduced oxidative products as BHT synthetic antioxidant and improved color (presented higher redness). Regarding to packing condition, no effect was observed on pink pepper and BHT treatments on lipid oxidation.	[43]
Clove extract (Tannins, sesquiterpenes, triterpenoids, eugenol, and eugenyl acetate)	0.10	Condensed aqueous extract added into the minced meat and was mixed thoroughly	Cooked beef patties	Packaged in a polyethylene pack and storage under refrigerated conditions for 10 days	The clove extract was more efficient to reduce lipid oxidation in cooked beef patties than BHT (0.02%) and ascorbic acid (0.05%); more efficient in preserving protein oxidation than BHT and similar to ascorbic acid at 10 days	[44]
Rosemary essential oil	0.015, 0.03, and 0.06	-	Different types of frankfurters: produced with tissues from Iberian pigs (IF) or white pigs (WF)	Storage in the dark for 60 days at 4 °C.	IF: 0.03, and 0.06% were effective to reduce lipid oxidation and 0.06% reduce hexanal formation. WF: 0.03, and 0.06% ppm increased lipid and protein oxidation	[45]
	0.2	Essential oil solution was applied on the fillets	Poultry fillets	Two different conditions: air-packaging and modified atmosphere	The combination of rosemary essential oil and modified-atmosphere packaging reduced the level of lipid oxidation.	[46]
Essential oils: clove; holy basil; cassia and thyme oil	Clove oil (0.25%), holy basil oil (0.125%), cassia oil (0.25%), and thyme oil (0.125%)	The essential oils were applied in thawed chicken meat before the sausage process	Fresh chicken sausage	Packed in a vacuum and stored at 18 ± 2 °C for 45 days	Clove oil was the most effective regarding reduce lipid oxidation. All essential oil presented antioxidant action and reduced discoloration during storage.	[47]

**Table 3 biomolecules-12-01506-t003:** Flowers, leaves, and vegetable residues as natural sources of antioxidants applied in meat products.

Source or Active Compound	Levels Added (%)	Adding Method	Sample	Storage Conditions	Effect	Ref.
Pitanga leaf extract (hydroxycinnamics and tyrosols)	0.02, 0.05 and 0.1	Powder extract	Pork burger	Polystyrene trays sealed with polyethylene film—modified atmosphere (80% O_2_ and 20% CO_2_); Refrigerated storage (2 ± 1 °C) under light to simulate supermarket conditions for 18 days	Reduce the count of bacteria when compared to the control and BHT at the end of storage. Improve color during the shelf-life (*a**), decreasing the discoloration; all concentrations inhibited lipid and protein oxidation during the storage and showed similar behavior to BHT	[5]
	0.025	Powder extract	Lamb burgers with fat replacement by chia oil emulsion	Package in a modified atmosphere (80% O_2_ and 20% CO_2_) and storage at refrigeration (2 ± 1 °C) under light	Most efficient to reduce lipid and protein oxidation than BHT (0.02%) and guarana seed extract (0.025%); presented higher antioxidant activity than BHT and similar guarana seed extract in 6 and 12 days of storage; avoided the formation of volatile compounds from lipid oxidation.	[1]
Wild thyme by-products extract (supercritical fluid extraction in different conditions) (Flavonoids, tannins, terpenoids)	7.5 and 15	Liquid extract	Ground pork patty	Packed in polypropylene trays and overwrapped with an oxygen-permeable polyvinyl chloride film and stored at 4 °C for 3 days.	TBARS values in pork patty with wild thyme by-products extract were lower than the control; protected proteins from oxidation; reduced discoloration during the storage time	[54]
Beer residue extract (BRE) (flavonoids—catechin, epicatechin and proanthocyanidins), some phenolic acids and bitter acids; Chestnut leaves extract (gallic and ellagic acid, rutin, quercetin, luteolin, epigallocatechin and kaempferol); peanut skin extract (proanthocyanidins)	2.0	Powder (BRE) and liquid extract (chestnut leave and peanut skin)	Spanish *salchichón* elaborated with encapsulated *n*-3 long chain fatty acids in konjac glucomannan matrix	**-**	BRE reduces protein oxidation and presented similar behavior to BHT; antioxidants were able to reduce the formation of aldehyde compounds and hexanal (volatiles compounds resulting from oxidation reactions)	[55]

## Data Availability

Not applicable.

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
