# Peer review of "Addition of Natural Extracts with Antioxidant Function to Preserve the Quality of Meat Products"

_biomolecules, 2022, doi:10.3390/biom12101506_

Round 1
Reviewer 1 Report
The manuscript focused to review the application of natural extracts derived from different plant sources in meat products and their influences in preserving the quality of these products. This review is concise and well-written and deserves to be suitable for publication in Biomolecules. However, I have some suggestions to improve the quality if this manuscript.
The major concerns:
I suggest to classify these natural antioxidants according to their chemical structures and clarify their antioxidant mechanisms. All tables should be provided with the adding methods of these natural antioxidants so that the readers can obtain the key information comprehensively.
Specific comments:
Title: This review mainly focused on the antioxidant capacity of different natural antioxidants. Microbiology plays an important role in the shelf-life of meat products. However, there are only a few of related antimicrobial properties of natural extracts introduced in this paper. Thus the shelf-life in this title should be reconsidered.
Lines 86-87: Please check the expression of malonaldehyde or MDA (1,3-propanedial).
Lines 91-93: Some other studies indicated different rancid thresholds for meat (2 or 2.28 mg MDA/kg, etc.). Please add these references.
Line 139:All “a*” should be italic.
Line 149: Please check all tables. How natural antioxidants affecting meat quality should be introduced rather than just using the word of “affect”. For example, 4% affect color, texture, and overall acceptance of the pork sausages. In addition, please unified the unit of additions.
Line 170: a better inhibited behavior on lipid oxidation?
Line 187: Please change “oxidation products” to “oxidative products”. Change to “consequently, inhibiting the increase in TBARS values during storage.
Line 214: Delete “(”.
Line 217: Whose membrane?
Line 228: Please change to “overall acceptance”.
Lines 277-278: Please change to “and its main bioactive compound has insecticidal and antioxidant properties”.
Lines 284, 285: What is the unit of these data?
Line 317: Please change to “in a vacuum packaging stored for 45 days”.
Line 357: Please describe these two extracts more precisely.
Line 369: Please change “between” to “among”.
Line 380: Please change to “reported that”.
Author Response
Reviewer #1 comments.
Comments and Suggestions for Authors
The manuscript focused to review the application of natural extracts derived from different plant sources in meat products and their influences in preserving the quality of these products. This review is concise and well-written and deserves to be suitable for publication in Biomolecules. However, I have some suggestions to improve the quality if this manuscript.
Response: The authors are thankful to the reviewers for their useful comments. Careful analysis of their suggestions has been made and the manuscript has been modified accordingly. We think that our paper has been improved due to their contribution and constructive suggestions. In the revised version, modifications are highlighted in red color so they could be easily distinguished from the original text.
The major concerns:
I suggest to classify these natural antioxidants according to their chemical structures and clarify their antioxidant mechanisms. All tables should be provided with the adding methods of these natural antioxidants so that the readers can obtain the key information comprehensively.
Response: Following the reviewer suggestion, the main chemical structures and the adding method were included in all natural antioxidants, which are in the Table 1, 2 and 3. The antioxidant mechanism was described in the text (line 124-140). So, all Tables were completed with adding methods.
Specific comments:
Title: This review mainly focused on the antioxidant capacity of different natural antioxidants. Microbiology plays an important role in the shelf-life of meat products. However, there are only a few of related antimicrobial properties of natural extracts introduced in this paper. Thus the shelf-life in this title should be reconsidered.
Response: The reviewer is right. Thus, following the reviewer´s indication, the title was changed, and “shelf-life” was removed. Additionally, the antimicrobial aspects were not covered, since there are several recent reviews that cover this aspect.
Lines 86-87: Please check the expression of malonaldehyde or MDA (1,3-propanedial).
Response: Thank you for your comment. The expression was changed.
Lines 91-93: Some other studies indicated different rancid thresholds for meat (2 or 2.28 mg MDA/kg, etc.). Please add these references.
Response: According to the reviewer´s indication, additional references were included. Moreover, not only additional references, but also additional information was included in the paragraph.
Line 139:All “a*” should be italic.
Response: The reviewer is right. This aspect was corrected throughout the text.
Line 149: Please check all tables. How natural antioxidants affecting meat quality should be introduced rather than just using the word of “affect”. For example, 4% affect color, texture, and overall acceptance of the pork sausages. In addition, please unified the unit of additions.
Response: In accordance with the reviewer indication, all Tables were checked and the unit was unified. The effects were changed following the reviewer´s suggestion, in order to improve this information.
Line 170: a better inhibited behavior on lipid oxidation?
Response: The sentence was corrected, in order to clarify this information.
Line 187: Please change “oxidation products” to “oxidative products”. Change to “consequently, inhibiting the increase in TBARS values during storage.
Response: The sentence and expressions were changed following the reviewer´s indications.
Line 214: Delete “(”.
Response: Thank you. The mistake was corrected.
Line 217: Whose membrane?
Response: The expression was clarified.
Line 228: Please change to “overall acceptance”.
Response: The sentence was corrected according to the reviewer´s indication.
Lines 277-278: Please change to “and its main bioactive compound has insecticidal and antioxidant properties”.
Response: The sentence was corrected according to the reviewer´s indication.
Lines 284, 285: What is the unit of these data?
Response: The units were corrected.
Line 317: Please change to “in a vacuum packaging stored for 45 days”.
Response: Following the reviewer´s suggestion, the sentence was corrected.
Line 357: Please describe these two extracts more precisely.
Response: According to the review indication, additional information about the extracts were included in the paragraph.
Line 369: Please change “between” to “among”.
Response: The reviewer is right. This mistake was corrected.
Line 380: Please change to “reported that”.
Response: The mistake was corrected.
Reviewer 2 Report
Some new references should be added to the review such as :
1. Novel non-specific lipid-transfer protein (TdLTP4) isolated from
durum wheat: Antimicrobial activities and anti-inflammatory properties
in lipopolysaccharide (LPS)-stimulated RAW 264.7 macrophages
2. Thiamine Demonstrates Bio-Preservative and Anti-Microbial Effects in
Minced Beef Meat Storage and Lipopolysaccharide (LPS)-Stimulated RAW
264.7 Macrophages
3. Microbiological Quality of Deer Meat Treated with Essential Oil
Litsea cubeba
Author Response
Reviewer #2 comments.
Comments and Suggestions for Authors
Some new references should be added to the review such as:
- Novel non-specific lipid-transfer protein (TdLTP4) isolated from durum wheat: Antimicrobial activities and anti-inflammatory properties in lipopolysaccharide (LPS)-stimulated RAW 264.7 macrophages
- Thiamine Demonstrates Bio-Preservative and Anti-Microbial Effects in Minced Beef Meat Storage and Lipopolysaccharide (LPS)-Stimulated RAW 264.7 Macrophages
- Microbiological Quality of Deer Meat Treated with Essential Oil Litsea cubeba
Response: Following the reviewer’s suggestion, the proposed references and additional information/discussion were included in the text.
Round 2
Reviewer 1 Report
In my opinion, this manuscript has been revised according to the reviewers's suggestions. Its quality is better now and can be considered to be accepted.